



# Estimating Ion Escape from Unmagnetized Planets

Mats Holmstrom

Swedish Institute of Space Physics, PO Box 812, SE-981 28 Kiruna, Sweden

**Correspondence:** Mats Holmstrom (matsh@irf.se)

**Abstract.** We propose a new method to estimate ion escape from unmagnetized planets that combines observations and models. Assuming that upstream solar wind conditions are known, a computer model of the interaction between the solar wind and the planet is executed for different ionospheric ion production rates. This results in different amounts of mass loading of the solar wind. Then we obtain the ion escape rate from the model run that best fit observations of the bow shock location. As an example of the method we estimate the heavy ion escape from Mars on 2015-03-01 to be $2 \cdot 10^{24}$ ions per second, using a hybrid plasma model and observations by MAVEN and Mars Express. This method enables studies of how escape depend on different parameters, and also escape rates during extreme solar wind conditions, applicable to studies of escape in the early solar system, and at exoplanets.

## 1 Introduction

Ion escape to space is important for the evolution of planetary atmospheres. Neutrals in the upper parts of the atmosphere can be ionized by, e.g., UV photons, charge exchange, and electron impacts. The newly created ion can then be energized by electric fields and transported away by the stellar wind, overcoming gravity, resulting in atmospheric loss.

For planets in our solar system, we can observe the present day escape of planetary ions by directly observing the ion flux near a planet. This is done using an ion detector on a spacecraft and gives us the flux of ions along the trajectory of the spacecraft. Since the flux of escaping ions is highly variable in time and location, accurately estimating the escape of ions can require observations over many years to get an average escape rate. To investigate how the escape rate of ions depend on different parameters, e.g., upstream solar wind conditions, is even more difficult due to the large amounts of observations needed to get sufficient statistics.

Another way to estimate ion escape rates is to use computer models of the solar wind interaction with a planet. An advantage of models compared to observations is that we at every instance get the full three-dimensional escape. In addition, there are no limitations in sensitivity, energy range or field of view as there is for observations. However, to accurately estimate ion escape requires that the model contains all important physical processes, in sufficient detail. It is questionable if this is the case at present.

Here we propose an alternative method of estimating ion escape. A method that uses both observations and computer models. Instead of directly observing the escaping ions, we use observations of other plasma quantities near the planet. Then we use a



parameterized model and find the set of model parameters that give the best fit between the model and the observations. The model escape rate for the best fit parameters then gives us an estimate of the ion escape rate.

This approach allow us to use data sets traditionally not used for ion escape estimates, e.g., magnetic field and electron observations. We can also estimate the escape rate from a very small set of observations, during one orbit of a space craft

around a planet, or during one flyby of a planet.

To illustrate this general method we estimate the escape rate of ions from Mars during one bow shock crossing of the MAVEN spacecraft (Jakosky et al., 2015) using magnetic field observations of the bow shock location, observations of the upstream solar wind conditions, and a hybrid plasma model. We also use Mars Express (MEX) (Barabash et al., 2007) observations of electrons to verify our findings.

The location and shape of the Martian bow shock has been the topic of many observational studies. Where is the bow shock located, and what are the controlling parameters? Recently Hall et al. (2016) found a seasonal dependence of the bow shock location, and later also a solar cycle dependence (Hall et al., 2019). The seasonal dependence should be due to the changing distance from the Sun of Mars. It is not straight forward to deduce if it is due to changing solar wind pressure or changing EUV insolation, since both scale in the same way with distance from the Sun. Also, both the solar wind and the EUV insolation

changes over a solar cycle. Regarding the effect of crustal magnetic fields on the bow shock location, Gruesbeck et al. (2018) find such a dependence, and a modeling study by Fang et al. (2017) finds that a large part of the variability in escape may be due to the crustal fields, reducing and enhancing escape, depending on the location. Regarding the shape of the bow shock, Vignes et al. (2002) found that it is furthest away from the planet in the hemisphere in the direction of the solar wind convective electric field.

However, it has been noted that, apart from the upstream solar wind conditions, the factor controlling the location of the shock is the amount of mass loading of the solar wind by ionospheric ions. This was noted for Venus by Alexander and Russell (1985), and later for Mars by Vignes et al. (2002) and Mazelle et al. (2004). The EUV flux, atmospheric state, ionospheric chemistry, magnetic anomalies, and similar factors will all affect the location of the bow shock, but only indirectly through the amount of mass loading. Therefore, given upstream solar wind conditions, we should be able to use the amount of mass

loading as a free parameter when modeling the location of the bow shock.

## 2  Method

The algorithm presented in this paper is general, and can be applied to any model of the interaction between Mars and the solar wind. To illustrate the method we here use a very simple hybrid model.

We now describe the hybrid plasma solver used, the adaptation for Mars, the parameters used; the observations of magnetic

field, ions and electrons used, and finally the algorithm to estimate ionospheric ion escape.





## 2.1 Hybrid model

In the hybrid approximation, ions are treated as particles, and electrons as a massless fluid. The trajectories of the ions are computed from the Lorentz force, given the electric and the magnetic fields. The electric field is

$$\mathbf{E} = \frac{1}{\rho_I}\left(-\mathbf{J}_I \times \mathbf{B} + \mathbf{J} \times \mathbf{B} - \nabla p_e\right) + \eta\mathbf{J}, \tag{1}$$

where $\rho_I$ is the ion charge density, $\mathbf{J}_I$ is the ion current density, $p_e$ is the electron pressure, and $\eta$ is the resistivity. The current is computed from, $\mathbf{J} = \mu_0^{-1}\nabla \times \mathbf{B}$, where $\mu_0 = 4\pi \cdot 10^{-7}$ is the magnetic constant.

Then Faraday's law is used to advance the magnetic field in time,

$$\frac{\partial \mathbf{B}}{\partial t} = -\nabla \times \mathbf{E}.$$

Further details on the hybrid model used here, the discretization, and the handling of vacuum regions can be found in Holmstrom et al. (2012).

## 2.2 Mars model

In the hybrid simulation domain, Mars is modeled as a resistive sphere, of radius $R$, centered at the origin, where all ions that hits the obstacle are removed from the simulation. The ionosphere is represented by the production of a single specie of ions according to an analytical Chapman ionospheric profile (Holmstrom and Wang, 2015).

We note that the ion production is a free parameter in such a model. This is in contrast to models that self consistently include ionospheric chemistry and a neutral atmosphere (Brecht et al., 2016). Usually this free parameter is seen as a limitation of the model, but here we use this as an advantage, to find a best fit to observations. This means that it is not important what the exact processes are in the ionosphere that produce the ions, or how they are transported to the top of the ionosphere.

Regarding the composition of escaping ionospheric ions. A study by Carlsson et al. (2006) based on Mars Express observations found flux ratios of $O_2^+/O^+ = 0.9$ and $CO_2^+/O^+ = 0.2$. Later Rojas-Castillo et al. (2019) estimated that $O_2^+/O^+ = 0.76$. Using MAVEN data, Inui et al. (2019) found a ratio of $O_2^+/O^+ = 1.2$, and that $CO_2^+$ contributed less than 10% to the total heavy ion flux. In summary, observations indicate that the flux of escaping $O^+$ and $O_2^+$ ions are of similar magnitude, while the $CO_2^+$ flux is not significant.

Since the code used here only handles one ionospheric ion specie, we make separate simulation runs using $O^+$ and $O_2^+$ to investigate the effects of composition on the ion escape rate estimates.

## 2.3 Model parameters

The coordinate system used is MSO coordinates with the solar wind flowing along the $-x$-axis, with density $n_{sw}$, speed $v_{sw}$ and temperature $T_{sw}$. The upstream interplanetary magnetic field is $\mathbf{B}_{sw}$. The computational grid has cubic cells of size $\Delta x$ and the time step is $\Delta t$. The computational domain is $-11000 \le x \le 10000$, $-33600 \le y, z \le 33600$ km. On the upstream boundary, after each timestep, we insert solar wind protons so that the number of particles per cell there is constant. In the $y$





and $z$ directions we have periodic boundary conditions. The produced ionospheric ions has a weight (how many real ions they represent) that is chosen such that the weight is similar to that of the solar wind protons. The model parameters and their values are listed in Table 1.

### 2.4 Observations

We use MAVEN magnetic field (Connerney et al., 2015) and ion (Halekas et al., 2017) observations to determine upstream conditions and bow shock location. The orbit is chosen such that the solar wind conditions are steady, so that the conditions should be unchanged while MAVEN is inside the bow shock, when we do not have observations of the solar wind. This also allow us to use a simulation that does not have time dependent upstream solar wind conditions. To verify our results we also use Mars Express (MEX) observations of electrons (Frahm et al., 2006) to locate bow shock crossings. The upstream solar

wind parameters are listed in Table 1.

### 2.5 Algorithm

Now we describe the algorithm for estimating the escape of ionospheric ions from observations of the upstream solar wind and the location of the bow shock. The algorithm is illustrated in Fig. 1.

1. We start with an observed state of the upstream solar wind: The magnetic field; the solar wind density, velocity and

temperature.

2. Then we make several runs of the hybrid model for these upstream solar wind conditions, using different ionospheric ion production rates.

3. We then find the simulation run that has a bow shock location that best correspond to the observed location. This could be done quantitatively, e.g., by a least square fit, but here we visually compare the simulations and the observations.

4. The escape for the best fit simulation run is then computed and this will be our estimate of the escape rate of ionospheric ions at the time of the bow shock observation.

### 3 Results

As an example of the proposed algorithm we performed ten simulations with the production rates $p_i = 0.1, 0.2, \ldots, 1.0 \, [\text{cm}^{-3}\text{s}^{-1}]$. In Fig. 2 we present a comparison of MAVEN observations and two hybrid runs with an $O^+$ ionosphere that were judged to

best fit the observations. We see that there is a fairly good agreement between the models and the observations at the bow shock and in the magnetosheath. Closer to the planet, in the induced magnetosphere, the agreement is however not so good. The magnetic field is much larger and more variable than in the model. This is not surprising since we have a model with a very simplified ionosphere, no magnetic anomalies, and low spatial resolution. Also, in the magnetosphere the proton velocities in





the model are much higher than observed. The proton density is however very small in this region. Also, the variability of $B_z$
in the magnetosheath is smaller in the models than observed.

As expected, and seen when comparing the two model runs, the location of the bow shock moves outward when the iono-
spheric ion production is increased, resulting in a larger mass loading.

The escape rate never reach a steady state due to intrinsic variability of the induced magnetosphere. So we determine the
escape rate by averaging the flux of ionospheric ions along $-x$ in the simulation domain downstream of $x = -5000$ km. This
is done at 10 s intervals from 200 s to 590 s, and then we average the escape over those times.

In Table 2 we show the results for the different simulation runs, numbered 1-4. For the best fit runs using $O^+$ the escape rate
is $2.0 \cdot 10^{24}$, while it is $1.5 \cdot 10^{24}$ s$^{-1}$ for $O_2^+$. Since we in reality has a mixture of the two ion species, the escape rate should
be between these values, and we can estimate the actual escape as $2 \cdot 10^{24}$ s$^{-1}$.

We can note how the bow shock location depends on the specie of the escaping ionospheric ions. A similar location is
obtained with 25% less escaping $O_2^+$ ions compared to $O^+$. It is not however directly proportional to the total mass of the
escaping ions, in that case we would expect a 50% reduction. So it is not only the amount of mass loading that determine the
bow shock location, the dynamics of the escaping ions is also important.

Looking at the escape in Table 2 for the same specie, but for different production rates, we see that the bow shock location
is very sensitive to the escape rate. Less than 1% variation in escape results in the change in bow shock location clearly visible
in Fig. 2. Another way to state this is that the escape rate is weakly dependent on the production rate.

To verify the location of the bow shock in the two best fit model runs, we also use Mars Express observations of the bow
shock in electron data. In Fig. 3 we plot the proton number density from the same two hybrid runs as in Fig. 2, but now along
the Mars Express orbit, together with the location of the bow shock crossings observed by MEX. The agreement is fairly good,
even if the observed bow shock is a few minutes earlier than seen in the model runs. This corresponds to a distance of a few
hundred kilometer, comparable to the simulation grid cell size. One reason for this could be that we have not used an aberrated
solar wind velocity (it flows along the $-x$ axis). This should result in a tilt of the whole magnetosphere and bow shock.

## 4 Discussion

For our example case we found an escape rate of $2 \cdot 10^{24}$ ions per second. This is in the range of recently published estimates
for the escape rate at Mars (Ramstad et al., 2015; Brain et al., 2015; Dong et al., 2017).

Note that there is no neutral H or O corona in the model. So those populations of exospheric pick-up ions are missing. That
means the mass loading due to the corona is missing in the simplified model we use. The total mass loading can be compensated
by more ions from the ionospheric source, but in reality the spatial distribution would be different. We also do not include any
alpha particles in the solar wind, that should have some effect on the solar wind interaction.

An assumption is that, given upstream condition, the mass loading determines the bow shock location. Although we find an
escape rate for a single orbit that match observed escape rates, this assumption needs to be tested in more detail. An ongoing



investigation is to apply the method to a large number of orbits and verify that the model estimated escape rates are consistent with observed escape rates.

The proposed method is directly applicable to unmagnetized planets. For magnetized planets the bow shock location is mainly determined by the upstream solar wind and the strength of the dipole field. Escape at magnetized planets occur in the

cusp regions, and how that affects the bow shock location, and if the presented method could be adapted to magnetized planets would need further investigation.

The bow shock location has been found to depend on the location of the magnetic anomalies relative to the solar wind flow (Fang et al., 2017). Is it because the fields "push out" the boundaries, or because the fields increase ion escape? The latter may not require crustal fields in the model used in our algorithm. Since the parameter we vary is the amount of ions near Mars

that is available to escape. If the crustal fields in a specific geometry enhance escape, this will be captured in the algorithm since the best fit bow shock will be further out, and if it depress escape the bow shock will be closer to the planet.

We used a hybrid plasma model in the algorithm. It would however be possible to use another type of plasma model, e.g., a magnetohydrodynamic (MHD) model, that can predict the location of the bow shock for different amounts of mass loading by ionospheric ions, given upstream solar wind conditions.

**5 Conclusions**

In the past, ion escape has been estimated either by computer models or from observations. Models have the problem that every physical process has to be present in the model. Observations suffer from variability, requiring the averaging of data over years, or even decades. The method proposed here uses a model and observations together. In that way we overcome the difficulties of each approach. We then get an estimate of the escape using just one observation. A model is used to estimate a global property

(ion escape) from a local observation (bow shock location). Since we use a model, there are no physical limitations in terms of energy coverage and field of view that observations have. In particular low energy escaping ions are difficult to observe.

This opens up the possibility of estimating escape during flybys of unmagnetized planets, in the past and in the future. It also allow for estimating escape twice per orbit given only a magnetometer and an ion detector. This enables detailed studies of how the escape depends on different parameters. Something that has been difficult in the past due to the years of observations

needed to collect enough statistics. It also makes possible the study of escape during transient events, like extreme solar wind conditions, something that is important for studies of escape in the past, and the escape at exoplanets.

*Data availability.* The MAVEN data used in this work, ion data from the SWIA instrument and magnetic field data from the MAG instrument, is available in NASA's Planetary Data System (PDS) at `https://pds-ppi.igpp.ucla.edu/search/view/?f=null&id=pds://PPI/maven.insitu.calibrated/data/2015/03`

The ASPERA-3 electron data used, from the ELS sensor, is available at `https://pds-ppi.igpp.ucla.edu/search/view/?id=pds://PPI/MEX-M-ASPERA3-3-RDR-ELS-EXT5-V1.0`



*Author contributions.* The author did all of the work.

*Competing interests.* The author declare that he has no conflict of interest.

*Acknowledgements.* Computing resources used in this work were provided by the Swedish National Infrastructure for Computing (SNIC) at
the High Performance Computing Center North (HPC2N), Umeå University, Sweden. The software used in this work was in part developed
by the DOE NNSA-ASC OASCR Flash Center at the University of Chicago.



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





**Table 1.** The parameters used for all simulation runs.

| Name | Symbol | value | unit |
|---|---|---|---|
| Inner boundary radius | $R$ | 3540 | km |
| Solar wind number density | $n_{\mathrm{sw}}$ | 2.4 | cm$^{-3}$ |
| Solar wind velocity | $u_{\mathrm{sw}}$ | 350 | km/s |
| Solar wind temperature | $T_{\mathrm{sw}}$ | $1.2 \cdot 10^5$ | K |
| Solar wind magnetic field | $\mathbf{B}_{sw}$ | $(-1, -2.7, -1)$ | nT |
| Plasma resistivity | $\eta$ | $5 \cdot 10^4$ | $\Omega$ m |
| Obstacle resistivity | | $7 \cdot 10^5$ | $\Omega$ m |
| Particles per cell | | 128 | |
| Weight of ionospheric ions | | $2.2 \cdot 10^{21}$ | |
| Height of max production | | 500 | km |
| Atmospheric scale height | | 250 | km |
| Cell size | $\Delta x$ | 350 | km |
| Time step | $\Delta t$ | 0.05 | s |





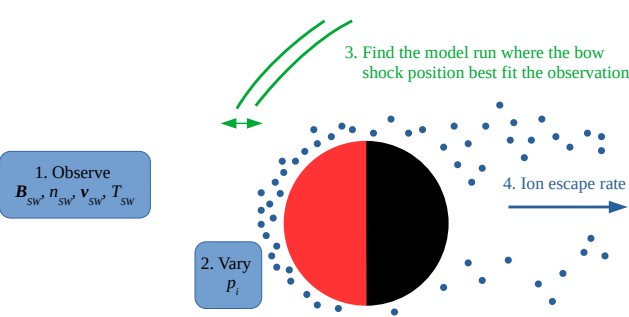

**Figure 1.** An illustration of the algorithm to estimate the ion escape rate. The Sun is to the left and Mars is the red and black disk. Using fixed upstream solar wind conditions from observations in the hybrid model, we vary the ionospheric heavy ion (blue dots) production rate for different simulation runs. The bow shock location (green lines) in each simulation run is then compared to the observed bow shock location. The estimated escape rate will then correspond to that of the simulation run that best fit the bow shock location.

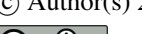



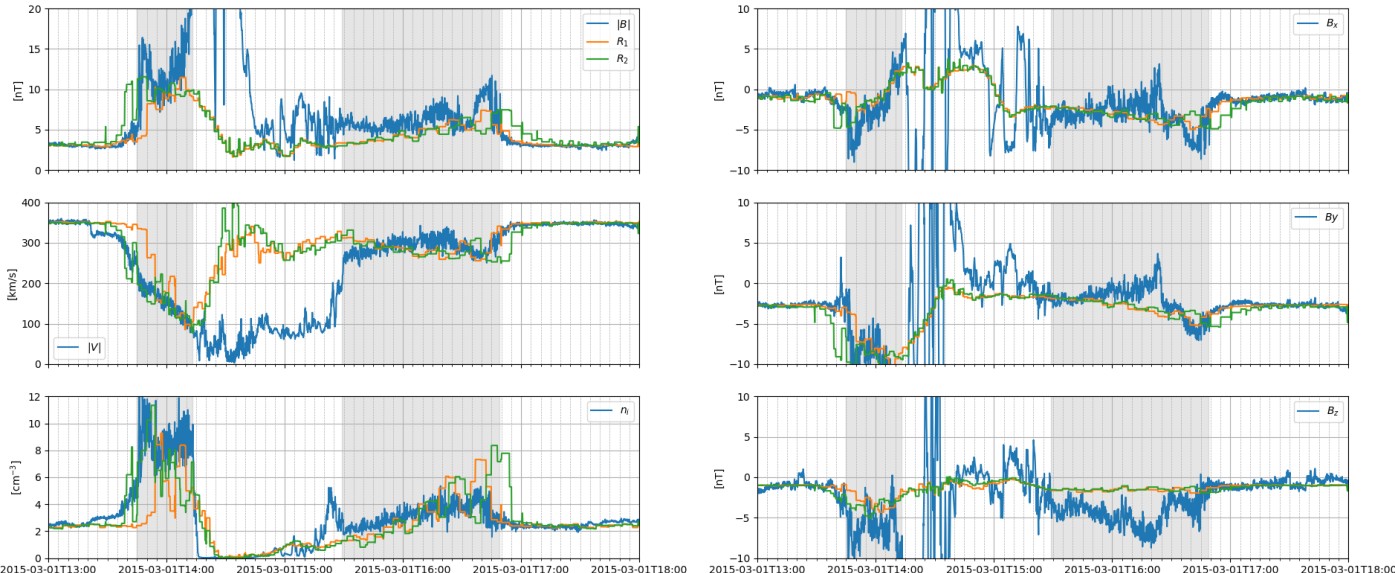

**Figure 2.** A comparison of MAVEN observations (blue) with two $O^+$ model runs ($R_1$ in orange and $R_2$ in green) at 490 s. In the left column, from the top, we have magnetic field magnitude, proton velocity, and proton number density. In the right column we have the three magnetic field components ($B_x$, $B_y$, and $B_z$) in MSO coordinates. Indicated in gray is also the location of the magnetosheath as seen from the observations. The induced magnetosphere is thus between the two gray regions.

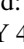



**Table 2.** The different best fit simulation runs, the ionospheric ion specie, the maximum ionospheric production rates, and the resulting escape rates.

| Run | 1 | 2 | 3 | 4 |
|---|---|---|---|---|
| Ionospheric specie | $O^+$ | $O^+$ | $O_2^+$ | $O_2^+$ |
| Production Rate, $p_i$ [cm$^{-3}$s$^{-1}$] | 0.4 | 0.5 | 0.4 | 0.5 |
| Escape Rate [$10^{24}$s$^{-1}$] | 1.985 | 1.989 | 1.452 | 1.463 |





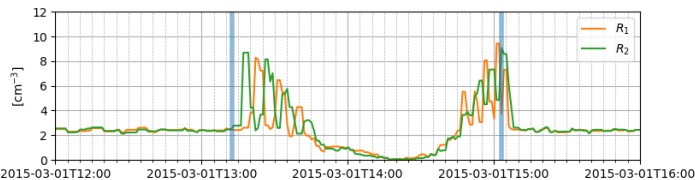

**Figure 3.** The proton number density along the MEX orbit for the two best fit simulation runs ($R_1$ in orange and $R_2$ in green), at 490 seconds of simulation time. The blue vertical lines show the two bow shock crossings seen in MEX electron data.