# Peer review of "Estimating Ion Escape from Unmagnetized Planets"

_Annales Geophysicae, 2021_

## Referee Comment (RC1)

The paper deals with testing of a novel approach to estimate ion escape from an unmagnetized planet. Quantifying of this important for atmospheric evolution feature is performed nowadays either by spacecraft in-situ measurements or with the numerical simulations. At the same time, none of these approaches can be considered as a sufficiently informative one, since the local (in space and time) spacecraft measurements are subject of strong fluctuations and they do not provide the global picture of ion escape from the planet; whereas numeric models may miss some important physics and give, therefore, erroneous predictions. The proposed approach tries to avoid of both kinds of such limitations, and it combines the in-situ observational data and computational modelling.

An idea of the method is quite straight forward. The author pays attention to the fact that besides of the solar wind conditions, the upstream location of the planetary bow shock is controlled by the mass loading of the solar wind by ionospheric ions. This means that the position of bow shock could be considered to certain extend as an indicator of the solar wind mass loading, which at the same time might be resulted by different physical mechanisms and drivers. This mass loading of the solar wind by ionospheric ions appears also a prerequisite of the ion escape in question. Therefore, instead of trying to reproduce the measured along the spacecraft trajectory ion fluxes, it is proposed to calculate the location of the bow shock for the given upstream solar wind conditions, using the ionospheric ion mass loading as a free parameter of the model. Then, the model run that gives the best correspondence between the location of the simulated bow shock and observations is used to calculate the ion escape rate. The position of real bow shock is judged from the direct measurements of magnetic field. The in-situ electron and ion data, and the upstream solar wind parameters used as the model input, are also taken from the corresponding observations outside the bow shock. A simple hybrid model with only one single-charged ion species was used.

This approach is illustrated to estimate the escape rate of ions from Mars, showing good agreement with other published estimates.

*I find the paper very interesting and worth to be published. The proposed method for the estimation of ion escape, in fact, remains still disputable in some aspects and generates questions, which I specify below. At the same time, this is a new method, which deserves further study, testing and broad community discussion. I expect that this paper would inspire all these processes, as well as my questions and comments (below) will be addressed by the author.*

Before going to details of the presented modelling, I would like to make a general comment. In fact, I have some difficulty with an overall picture of the model scenario, which involves 1) the solar wind inflow, 2) the ionospheric source of mass loading ions at the boundary of conducting obstacle, and 3) the escaping ion flux. Is a kind of a steady-state conservation condition assumed for the species here? If so, then all injected ionospheric ions have to be continuously removed, which means that part of them is blown away with the wind, i.e. appears the escaping ions of interest, and another part is transported down to the planet and disappears at the surface boundary. Since continuous grow of the ion population around the planet (due to the operation of the ionospheric source), as well as degeneration of the ion environment because of a strong ion escape do not seem realistic, I suppose the above mentioned dynamical balance between the sources and sinks to be the only reasonable state. This, however, means that the escape rate of ions cannot exceed the ion production at the ionospheric boundary. Taking the used in paper production rate density of 0.5 [cm^-3 s^-1] equally distributed in the spherical shell of unit thickness with the radius equal to the radius of inner boundary of the simulation domain (3540 km), one can obtain the total ion production rate of ~7.9 x 10^17 [s^-1]. It is essentially less than the ion escape rate of ~2 x 10^24 [s^-1], obtained in the simulations. So, more ions are escaping than produced. It looks inconsistent. There are however no details regarding the structure of the prescribed ionospheric ion source. If we take it equally distributed over the whole inner boundary sphere of 3540 km radius, then the total ion production rate becomes to be ~9.3 x 10^25 [s^-1], which is higher than the simulated ion escape rate (~2 x 10^24 [s^-1]), and the inconsistency is solved. However, such a source inside the conducting planetary obstacle has to be reasonably justified.

In any case, more details about the prescribed ion source are needed. Do the newly appeared particles have some initial velocity? How the ion source is distributed in the volume around/over the planet (equal distribution; location on the day side; else)?

May be instead of prescribing the ion production rate $p_i$ [cm^-3 s^-1] in some volume, it is better and more realistic just to fix the ion density $n_i$ [cm^-3] at the inner boundary, so that after each time-step the amount of ions in each boundary cell is upgraded to a given constant value?

Below follow my *questions and comments* regarding more specific aspects of the applied model and its presentation.

**(1)** The used hybrid model is indeed very simple. In fact, it does not include the neutral species and completely ignores the effect of the charge-exchanged particles' pick-up. There is a statement on that in line 140 (in Discussion section), but I would recommend to address all such simplifications and assumptions in Section 2, where the method is described and the model is introduced.

**(2)** The model uses only one single-charged ion species which in course of the study is taken to be either O+ or O2+. At the same time according to MAVEN data (referred also in the paper), both these ion species are almost equally present in the escaping ion flux, so their separate treatment, when only one of two is considered and another is completely ignored, has to be justified.

**(3)** The presence of term with resistivity in equation (1) for the electric field means that the momentum exchange between electrons and at least protons, due to Coulomb collisions, is taken into account. This in its turn means that electrons are not completely massless, as written in Line 57. This kind of approximation corresponds to neglecting of the electron inertia, i.e., taking $m_e \, dV_e/dt = 0$ in the corresponding momentum equation. In that respect a question is how the simulated heavy ions (O+ and O2+) take part in the momentum exchange and contribute to the electric current?

**(4)** How the quasi-neutrality is insured in course of the simulations, given there is a source of ions, operating at the ionospheric boundary? Is the charge of injected ions compensated by the same amount of injected electrons? The same question appears regarding the statement in Line 85, saying "On the upstream boundary, after each timestep, we insert solar wind protons…". Simple inserting of protons would increase an uncompensated positive charge.

**(5)** Production of ions, prescribed at the inner boundary of the simulation domain, would change the conductivity of plasma inside the domain. Is this effect taken into account in the model?

**(6)** How the electric currents induced on the planet (modelled as a resistive sphere), are taken into consideration?

**(7)** The value of plasma resistivity (5 x 10^4 [m Ohm]) in Table 1 is too high and inconsistent with that of the plasma with temperature 1.2 x 10^5 K (according Table 1). Defined as the reversed Spritzer's conductivity: $1/(10^{-3} T[K]^{(3/2)})$, the resistivity of such plasma should be 2.4 x 10^-5 [m Ohm]. I notice, however, that the value of plasma conductivity, calculated for T=1.2 x 10^5 K with the Spritzer's formula is 4,15 x 10^4 [Sm m^-1], i.e. numerically quite close to the figure in Table 1. Therefore I am inclined to think that instead of the plasma resistivity, the conductivity value is provided there by mistake. I hope however that the model operates with correct numbers and expect that the author will check and confirm it.

**(8)** It is also necessary that the resistivity value (7 x 10^5 [m Ohm]), taken for the planetary obstacle (Table 1), is justified with some physics reasoning. Is it attributed with the conductivity of planetary body, or with the ionospheric plasma? In the last case it seems to be too high.

**(9)** The atmospheric scale height is usually defined in the hydrostatic approximation of medium, and it depends on the temperature of fluid. Since in hybrid model applied in the paper the ions are treated kinetically, their temperature is undefined, and the scale height makes no sense for them. Then, it would be good to specify in the text for which species, and how, the atmospheric scale height of 250 km, provided in Table 1, is defined and how it is used in the model?

**(10)** Altogether, it seems that the used model cannot reproduce well the induced magnetosphere, generated by the inductive currents running in the conducting obstacle and surrounding plasma. As result, the missing magnetic pressure of the inner magnetosphere is compensated by the particles' ram pressure with unrealistically high velocity (mid panel in left column in Figure 2). In view of the lack of details regarding the account of the obstacle's conductivity and the integration of surrounding induced fields and magnetospheric current system, it is difficult to judge on the reason of such failure of the model.

**(11)** The difference between the observed bow shock and its modelled prototype along the Mars Express orbit in Fig.3 is estimated as a few hundred kilometers and attributed to the spatial resolution of the numerical model. However, as it can be seen in Figs.2 an 3, the difference between the simulation runs $R_1$ and $R_2$ is also of the same scale. If this difference is indeed comparable with the grid resolution, then the question is to which extend the model runs $R_1$ and $R_2$ can be treated as different ones?

Below are several my suggestions for phrasing, corrections of some sentences and their meaning discussion.

Lines 67-68: "…ions that hits the obstacle…" → "…ions that hit the obstacle…"

Line 76: "found a ratio of O2+/O+…" → "found a flux ratio of O2+/O+…"

Line 79: "…one ionospheric ion specie…"…one single-charged ionospheric ion species…"

Line 86: "…ionospheric ions has…" → "…ionospheric ions have…"

Line 109: "…two hybrid runs with an O+ ionosphere that…" → "…two hybrid runs ($R_1$ and $R_2$) with an O+ ionosphere and different ion production rates that…"

Lines 118-120: "So we determine the escape rate by averaging the flux of ionospheric ions along –x in the simulation domain downstream of x = −5000 km. This is done at 10 s intervals from 200 s to 590 s, and then we average the escape over those times." → "So, we determine the escape rate by averaging the flux of ionospheric ions along x in the simulation domain at x = −5000 km. This is done by averaging of the flux values calculated between 200 s and 590 s with the time step of 10 s."

Lines 121: "In Table 2 we show the results for the different simulation runs, numbered 1-4." → "In Table 2 we show the results of simulation runs $R_1 – R_4$, performed for O+ and O2+ ionosphere with different ion production rates."

Lines 121-122: "…the escape rate is 2.0 x 10^24, while it is..." → "…the approximate escape rate is 2.0 x 10^24 s^-1, while it is..."

Line 122: "Since we in reality has a mixture of…" → "Since in reality we have a mixture of…"

Line 128: "Looking at the escape in Table 2 for the same specie, but for different production rates, we see that the bow shock location is very sensitive to the escape rate." There is no information in Table 2 about the location of bow shock. So, it is not easy to see the relation between the ion escape rate

and bow shock location just looking at Table 2. Adding of a column with the corresponding bow shock distances would be helpful in that respect. I am also inclined to regard the value of ion escape rate as a result of specific position of the bow shock, achieved in the simulations for a given ion production rate, and not vice versa. In that respect, the sentence in Line 129 ("Less than 1% variation in escape results in the change in bow shock location clearly visible in Fig. 2") sounds strange to me. More logical would be to consider the modelled bow shock location dependent on the ion production rate, which is the model free parameter, taking then the simulated ion escape rate as just another model prediction, i.e. a function of the input parameter set.

---

## Author Response (AR1)

**> RC1: 'Comment on angeo-2021-40', Anonymous Referee #1, 17 Aug 2021**

*Italics = Anonymous Referee #1 text*
Roman = Author text

*> The paper deals with testing of a novel approach to estimate ion escape from an unmagnetized planet. Quantifying of this important for atmospheric evolution feature is performed nowadays either by spacecraft in-situ measurements or with the numerical simulations. At the same time, none of these approaches can be considered as a sufficiently informative one, since the local (in space and time) spacecraft measurements are subject of strong fluctuations and they do not provide the global picture of ion escape from the planet; whereas numeric models may miss some important physics and give, therefore, erroneous predictions. The proposed approach tries to avoid of both kinds of such limitations, and it combines the in-situ observational data and computational modelling.*

*> An idea of the method is quite straight forward. The author pays attention to the fact that besides of the solar wind conditions, the upstream location of the planetary bow shock is controlled by the mass loading of the solar wind by ionospheric ions. This means that the position of bow shock could be considered to certain extend as an indicator of the solar wind mass loading, which at the same time might be resulted by different physical mechanisms and drivers. This mass loading of the solar wind by ionospheric ions appears also a prerequisite of the ion escape in question. Therefore, instead of trying to reproduce the measured along the spacecraft trajectory ion fluxes, it is proposed to calculate the location of the bow shock for the given upstream solar wind conditions, using the ionospheric ion mass loading as a free parameter of the model. Then, the model run that gives the best correspondence between the location of the simulated bow shock and observations is used to calculate the ion escape rate. The position of real bow shock is judged from the direct measurements of magnetic field. The in-situ electron and ion data, and the upstream solar wind parameters used as the model input, are also taken from the corresponding observations outside the bow shock. A simple hybrid model with only one single-charged ion species was used.*

*> This approach is illustrated to estimate the escape rate of ions from Mars, showing good agreement with other published estimates.*

*> I find the paper very interesting and worth to be published. The proposed method for the estimation of ion escape, in fact, remains still disputable in some aspects and generates questions, which I specify below. At the same time, this is a new method, which deserves further study, testing and broad community discussion. I expect that this paper would inspire all these processes, as well as my questions and comments (below) will be addressed by the author.*
* * *
I thank the reviewer for insightsful and constructive comments that I think has improved the paper. Please find my replies below to the issues brought up. The content of my replies is implemented in the revised paper.

*> Before going to details of the presented modelling, I would like to make a general comment. In fact, I have some difficulty with an overall picture of the model scenario, which involves 1) the solar wind inflow, 2) the ionospheric source of mass loading ions at the boundary of conducting obstacle, and 3) the escaping ion flux. Is a kind of a steady-state conservation condition assumed for the species here? If so, then all injected ionospheric ions have to be continuously removed, which means that part of them is blown away with the wind, i.e. appears the escaping ions of interest, and another part is transported down to the planet and disappears at the surface*

*boundary. Since continuous grow of the ion population around the planet (due to the operation of the ionospheric source), as well as degeneration of the ion environment because of a strong ion escape do not seem realistic, I suppose the above mentioned dynamical balance between the sources and sinks to be the only reasonable state. This, however, means that the escape rate of ions cannot exceed the ion production at the ionospheric boundary. Taking the used in paper production rate density of 0.5 [cm^-3 s^-1] equally distributed in the spherical shell of unit thickness with the radius equal to the radius of inner boundary of the simulation domain (3540 km), one can obtain the total ion production rate of ~7.9 x 10^17 [s^-1]. It is essentially less than the ion escape rate of ~2 x 10^24 [s^-1], obtained in the simulations. So, more ions are escaping than produced. It looks inconsistent. There are however no details regarding the structure of the prescribed ionospheric ion source. If we take it equally distributed over the whole inner boundary sphere of 3540 km radius, then the total ion production rate becomes to be ~9.3 x 10^25 [s^-1], which is higher than the simulated ion escape rate (~2 x 10^24 [s^-1]), and the inconsistency is solved. However, such a source inside the conducting planetary obstacle has to be reasonably justified.*

It is correct that a balance is assumed between ion production and loss (through escape and through ions removed that cross the inner boundary) on the timescale of the simulation running time in this paper (490 seconds). So it is also correct that the escape rate cannot be larger than the production rate, averaged over the time of a simulation. Variability on a shorter time scale is possible due to the inherent variability of the simulation and the physical system. This is the reason for computing the escape rate by averaging as described on Line 118.

Regarding the ionospheric source. What is assumed is an analytical Chapman ionospheric profile (Line 69). A reference to an earlier publication is given that provide more details, but we now incorporate some of that material in this paper.

To get the total production rate of ions one would have to integrate the above mentioned Chapman profile.

*> In any case, more details about the prescribed ion source are needed. Do the newly appeared particles have some initial velocity? How the ion source is distributed in the volume around/over the planet (equal distribution; location on the day side; else)?*

As noted above, this is now described in the paper.

*> May be instead of prescribing the ion production rate p_i [cm^-3 s^-1] in some volume, it is better and more realistic just to fix the ion density n_i [cm^-3] at the inner boundary, so that after each time-step the amount of ions in each boundary cell is upgraded to a given constant value?*

Yes, that would be one possibility. Actually, the idea behind the proposed method is that the results should not be sensitive to the exact way the mass loading ions are produced near the inner boundary since the total amount of produced ions is a free parameter that we vary until we find a good fit with observations of the bow shock location. That the results are not sensitive to the exact details of the ionospheric source would be verified in a longer follow up paper.

*> Below follow my questions and comments regarding more specific aspects of the applied model and its presentation.*

*> (1) The used hybrid model is indeed very simple. In fact, it does not include the neutral species and completely ignores the effect of the charge-exchanged particles' pick-up. There is a statement on that in line 140 (in Discussion section), but I would recommend to address all such simplifications and assumptions in Section 2, where the method is described and the model is introduced.*

This discussion is now moved to Section 2.

> *(2) The model uses only one single-charged ion species which in course of the study is taken to be either O+ or O2+. At the same time according to MAVEN data (referred also in the paper), both these ion species are almost equally present in the escaping ion flux, so their separate treatment, when only one of two is considered and another is completely ignored, has to be justified.*

This is indeed a very simplified view.  However, as stated on Line 122, I think it is reasonable to assume that the escape rate of a more realistic ionosphere with O+, O2+, and CO2+, will lay in between the single specie rates.  This will be investigated in a longer follow up paper.

> *(3) The presence of term with resistivity in equation (1) for the electric field means that the momentum exchange between electrons and at least protons, due to Coulomb collisions, is taken into account. This in its turn means that electrons are not completely massless, as written in Line 57. This kind of approximation corresponds to neglecting of the electron inertia, i.e., taking $m_e dV_e/dt = 0$ in the corresponding momentum equation. In that respect a question is how the simulated heavy ions (O+ and O2+) take part in the momentum exchange and contribute to the electric current?*

Resistivity in a hybrid simulation satisfies a diffusion equation.  It can be viewed as representing electron physics and wave particle interactions that are not captured by the hybrid method.  The resistivity affects the magnetic field, that in turn will affect the motion of the ions (and the current that they represent).   Resistivity smooths out the magnetic fields, and some resistivity is usually required in hybrid simulations to avoid numerical instabilities.

> *(4) How the quasi-neutrality is insured in course of the simulations, given there is a source of ions, operating at the ionospheric boundary? Is the charge of injected ions compensated by the same amount of injected electrons? The same question appears regarding the statement in Line 85, saying "On the upstream boundary, after each timestep, we insert solar wind protons…". Simple inserting of protons would increase an uncompensated positive charge.*

Exact charge neutrality is a fundamental assumption of the hybrid model.  We can view this as the production of an exact equal amount of electron charge wherever ions are produced.

> *(5) Production of ions, prescribed at the inner boundary of the simulation domain, would change the conductivity of plasma inside the domain. Is this effect taken into account in the model?*

Yes, the change in plasma conductivity (and resulting currents) is self consistently accounted for by the ions themselves and the corresponding mass less electron fluid.

> *(6) How the electric currents induced on the planet (modelled as a resistive sphere), are taken into consideration?*

Even though the planet is resistive, some currents will be induced.

> *(7) The value of plasma resistivity ($5 \times 10^4$ [m Ohm]) in Table 1 is too high and inconsistent with that of the plasma with temperature $1.2 \times 10^5$ K (according Table 1). Defined as the reversed Spritzer's conductivity: $1/(10^{-3} T[K]^{(3/2)})$, the resistivity of such plasma should be $2.4 \times 10^{-5}$ [m Ohm]. I notice, however, that the value of plasma conductivity, calculated for $T=1.2 \times 10^5$ K with the Spritzer's formula is $4,15 \times 10^4$ [Sm m$^{-1}$], i.e. numerically quite close to the figure in Table 1. Therefore I am inclined to think that instead of the plasma resistivity, the conductivity value*

*is provided there by mistake. I hope however that the model operates with correct numbers and expect that the author will check and confirm it.*

As mentioned earlier, the resistivity in a hybrid model should be viewed as smoothing of the magnetic fields.  Representing the smaller scale electron physics and wave particle interactions that are not captured by the hybrid model.  Also, the amount of resistivity has to be chosen to ensure numerical stability.

*> (8) It is also necessary that the resistivity value (7 x 10^5 [m Ohm]), taken for the planetary obstacle (Table 1), is justified with some physics reasoning. Is it attributed with the conductivity of planetary body, or with the ionospheric plasma? In the last case it seems to be too high.*

This is the resistivity of the planetary body.

*> (9) The atmospheric scale height is usually defined in the hydrostatic approximation of medium, and it depends on the temperature of fluid. Since in hybrid model applied in the paper the ions are treated kinetically, their temperature is undefined, and the scale height makes no sense for them. Then, it would be good to specify in the text for which species, and how, the atmospheric scale height of 250 km, provided in Table 1, is defined and how it is used in the model?*

This applies to the analytical Chapman ionospheric profile mentioned earlier.  The produced ions are then randomly drawn from this profile, as now described in more detail in Section 2.

*> (10) Altogether, it seems that the used model cannot reproduce well the induced magnetosphere, generated by the inductive currents running in the conducting obstacle and surrounding plasma. As result, the missing magnetic pressure of the inner magnetosphere is compensated by the particles' ram pressure with unrealistically high velocity (mid panel in left column in Figure 2). In view of the lack of details regarding the account of the obstacle's conductivity and the integration of surrounding induced fields and magnetospheric current system, it is difficult to judge on the reason of such failure of the model.*

The proton velocity is indeed much higher in the model than in observations inside the induced magnetosphere.  However, the proton density is very small, close to zero, in much of this region both as observed and in the simulations.  Near the exit from the induced magnetosphere the observed density is larger than in the simulations, resulting in a similar dynamic pressure.  This is now discussed in the revised paper.
One can also note that the obstacle in the model is not completely resistive, even if the resistivity is large in the presented simulations (Table 1).  So there will be some induced currents in the obstacle.

*> (11) The difference between the observed bow shock and its modelled prototype along the Mars Express orbit in Fig.3 is estimated as a few hundred kilometers and attributed to the spatial resolution of the numerical model. However, as it can be seen in Figs.2 an 3, the difference between the simulation runs R_1 and R_2 is also of the same scale. If this difference is indeed comparable with the grid resolution, then the question is to which extend the model runs R_1 and R_2 can be treated as different ones?*

The two model runs can be seen as bracketing the observation, and thus both corresponding escape rates can be seen as the result.  In Table 2 the number of significant digits in the escape rates are such that they convey this uncertainty.

*> Below are several my suggestions for phrasing, corrections of some sentences and their meaning discussion.*

I thank the reviewer for a careful reading! Most of the suggestions below has been implemented in the revised paper.

> *Lines 67-68: "…ions that hits the obstacle…" à "…ions that hit the obstacle…"*
Corrected

> *Line 76: "found a ratio of O2+/O+…" à "found a flux ratio of O2+/O+…"*
Corrected

> *Line 79: "…one ionospheric ion specie…"…one single-charged ionospheric ion species…"*
Corrected

> *Line 86: "…ionospheric ions has…" à "…ionospheric ions have…"*
Corrected

> *Line 109: "…two hybrid runs with an O+ ionosphere that…" à "…two hybrid runs (R_1 and R_2) with an O+ ionosphere and different ion production rates that…"*
Changed

> *Lines 118-120: "So we determine the escape rate by averaging the flux of ionospheric ions along −x in the simulation domain downstream of x = −5000 km. This is done at 10 s intervals from 200 s to 590 s, and then we average the escape over those times." à "So, we determine the escape rate by averaging the flux of ionospheric ions along x in the simulation domain at x = −5000 km. This is done by averaging of the flux values calculated between 200 s and 590 s with the time step of 10 s."*

The flux along the x-axis would be a negative number. To get the positive escape rate, I prefer to call it "flux along -x".
The flux is actually averaged from x = -5000 km to the outflow boundary of the domain.
I have changed the text of the paragraph so that should be more clear.

> *Lines 121: "In Table 2 we show the results for the different simulation runs, numbered 1-4." à "In Table 2 we show the results of simulation runs R_1 − R_4, performed for O+ and O2+ ionosphere with different ion production rates."*
Changed

> *Lines 121-122: "…the escape rate is 2.0 x 10^24, while it is..." à "…the approximate escape rate is 2.0 x 10^24 s^-1, while it is..."*
Changed

> *Line 122: "Since we in reality has a mixture of…" à "Since in reality we have a mixture of…"*
Changed

> *Line 128: "Looking at the escape in Table 2 for the same specie, but for different production rates, we see that the bow shock location is very sensitive to the escape rate." There is no information in Table 2 about the location of bow shock. So, it is not easy to see the relation between the ion escape rate and bow shock location just looking at Table 2. Adding of a column with the corresponding bow shock distances would be helpful in that respect. I am also inclined to regard the value of ion escape rate as a result of specific position of the bow shock, achieved in the simulations for a given ion production rate, and not vice versa. In that respect, the sentence in Line 129 ("Less than 1% variation in escape results in the change in bow shock location clearly visible in Fig. 2") sounds strange to me. More logical would be to consider the modelled bow shock*

*location dependent on the ion production rate, which is the model free parameter, taking then the simulated ion escape rate as just another model prediction, i.e. a function of the input parameter set.*

This is a very good point! The description was not clear. The ion production rate affects the bow shock position and the escape rate, not the other way around. The text has been rewritten so that it only discuss ion production.